# High Sensitivity Resists for EUV Lithography: A Review of Material Design Strategies and Performance Results

**DOI:** 10.3390/nano10081593

**Published:** 2020-08-14

**Authors:** Theodore Manouras, Panagiotis Argitis

**Affiliations:** 1Department of Materials Science and Technology, University of Crete, 70013 Heraklion, Greece; 2Institute of Electronic Structure and Laser, Foundation for Research and Technology-Hellas, 70013 Heraklion, Greece; 3Institute of Nanoscience and Nanotechnology, NCSR “Demokritos”, 15310 Athens, Greece

**Keywords:** EUV lithography, chemically amplified resists, inorganic resists, main chain scission resists, electron-induced chemistry

## Abstract

The need for decreasing semiconductor device critical dimensions at feature sizes below the 20 nm resolution limit has led the semiconductor industry to adopt extreme ultra violet (EUV) lithography with exposure at 13.5 nm as the main next generation lithographic technology. The broad consensus on this direction has triggered a dramatic increase of interest on resist materials of high sensitivity especially designed for use in the EUV spectral region in order to meet the strict requirements needed for overcoming the source brightness issues and securing the cost efficiency of the technology. To this direction both fundamental studies on the radiation induced chemistry in this spectral area and a plethora of new ideas targeting at the design of new highly sensitive and top performing resists have been proposed. Besides the traditional areas of acid-catalyzed chemically amplified resists and the resists based on polymer backbone breaking new unconventional ideas have been proposed based on the insertion of metal compounds or compounds of other highly absorbing at EUV atoms in the resist formulations. These last developments are reviewed here. Since the effort targets to a new understanding of electron-induced chemical reactions that dominate the resist performance in this region these last developments may lead to unprecedented changes in lithographic technology but can also strongly affect other scientific areas where electron-induced chemistry plays a critical role.

## 1. Introduction—Nanostructure Formation in Semiconductor Lithography

The standard methodology for fabricating miniaturized devices with critical dimensions in the micro- and nano- domain in the semiconductor industry has been photolithography, using photosensitive polymeric mainly materials, known as resists, for imaging, accompanied with pattern transfer to the substrate of interest with plasma etching. The uninterrupted patterning technology evolution from the 1960s to the last decade allowed the miniaturization of critical device dimensions from a few microns to the sub 25 nm domain allowing the semiconductor technology progress according to the well-known Moore’s law [1,2,3]. Although this impressive technology development has been mainly achieved by adopting reduced wavelength of the imaging radiation, in the last two decades additional technology breakthroughs such as immersion and double exposure patterning allowed the shrinkage of device dimensions, while keeping the wavelength at 193 nm [4]. In addition, options like directed self-assembly [5,6,7,8], e-beam lithography [9,10,11,12], maskless techniques [13,14], and nanoimprint technology [15,16] have been heavily explored. At this period, it seems that the implementation of exposure at reduced wavelength, and in particular at 13.5 nm, is the choice of the big semiconductor industries for the future device technology. Extreme ultraviolet (EUV) lithography using 13.5 nm wavelength exposure is expected to be the main industrial option for pushing further the resolution limit in sub 20 nm region. However, EUV sources have limited power making the improvement in resist sensitivity a high-importance issue to fulfill the throughput requirements for high volume manufacturing while maintaining pattern fidelity and uniformity.

Reviews on EUV resists have been published in 2017 summarizing important developments [17,18]. Nevertheless, dramatic changes are under way in the field after consensus in the semiconductor industry community has been reached to accept EUV as the main next generation lithographic option. New unconventional materials are introduced and fundamental studies are under way enabled by improvements in the scientific understanding of the EUV radiation interactions with the resist materials. In this context a very recent review [19], focuses especially on inorganic photoresists covering materials that have been introduced to address the challenges encountered in EUV lithography.

In the present review we will discuss primarily the main factors that influence the resist sensitivity at EUV and we will give priority to promising materials for achieving the high sensitivity demanded. We will focus mostly on the developments during the last years aiming at new generations of materials and on the scientific support from fundamental studies toward technology optimization. Following this approach we discuss first in two separate sections the main characteristics of EUV technology and the issues related to materials absorption at this spectral region and then we review the chemical approaches for high sensitivity EUV resists and the results reported so far on resist materials of advanced performance.

## 2. EUV Introduction and Main Technical Challenges

The critical dimension (CD) of a projection imaging system such as a lithographic scanner is given by the fundamental equation governing the resolution: CD = kλ/NA, where λ is the wavelength of the light source and NA is the numerical aperture of the imaging optics. The value of κ is depending on the quality of the imaging optics, the way the mask is illuminated, and the mask pattern itself. As it is obvious from the latter equation, the critical dimension value is proportional to the wavelength of the radiation applied to the photosensitive materials.

The choice of 13.5 nm, or EUV as it is commonly referred to, as the imaging radiation wavelength after the 193 nm has a long history of scientific investigations, technology breakthroughs and controversies that lasted more than two decades [20]. A review of the first years of EUV lithography has been presented in [21], where also the successful demonstration of printing for the first time 19 nm dense lines/spaces with optical lithography using an interferometric Lloyd’s mirror maskless set-up is discussed [22]. The current patterning challenges at EUV aiming at a viable industrial technology have been also presented in [21] and more recently in [23,24]. One attractive characteristic of the EUV radiation at 13.5 nm is that it can be combined with imaging technology based on reflective optics, a technology that has reached an acceptable maturity level. This imaging technology, based on multilayer mirrors composed of technologically acceptable materials, Mo and Si, allows imaging of the mask pattern to the wafer. The reflectivity of such multilayers is discussed for instance in [25], and a characteristic graph from this paper is depicted in Figure 1a, where also emission spectra of different alternative sources are given. This graph is also based on data from reference [26]. The reflective optics used in EUV technology is apparently more complicated than the refractive optics used at 193 nm and longer wavelengths, see for instance a scheme of a projection system depicted in Figure 1b, adopted by ref [17], which as stated there is based on a previous presentation [27]. In addition, new technological challenges in mask fabrication had to be addressed, but the technology nowadays has been proved capable for effective radiation imaging. The successful development of this imaging technology for 13.5 nm radiation has played a very important role for the choice of this particular wavelength for next generation lithography.

One very challenging issue for the development of industrially competitive EUV lithography technology has been the brightness of the sources that could be used for generation of radiation at 13.5 nm. Plasma discharge approaches and approaches based on laser induced radiation through focusing on specific material targets were explored by many groups toward the development of efficient sources. Today the approach of choice seems to be laser-induced radiation generation by focusing on liquid Sn droplets. Sources based on this approach have been incorporated in the commercial EUV lithography exposure tools available today, as is also shown in the scheme of Figure 1b. The brightness of these sources has been greatly improved over the past decade and played a significant role for the decision to adopt EUV lithography as the next production technology made by a number of big semiconductor manufacturers [28]. Nevertheless, the throughput of the exposure tool remains relatively low compared with older lithographic technologies and for this reason there is an urgent demand for high sensitivity resists that can help the technology to achieve the desired cost efficiency. In addition, we should always keep in mind that the EUV resists must also provide advanced performance characteristics to be suitable for sub 20 nm patterning with industrial standards, related primarily to pattern quality and process demands.

This demand for high sensitivity EUV resists caused a real awakening of the research in the resists field. At this point it should be emphasized that the research effort in the resist development area has been rather slow for a period of more than 10 years (schematically extending from about 2005 to 2015). Indeed, after the abandonment of the option for 157 nm lithography, which had provoked an intense resist development effort, the resist material research was not intense since the 193 nm resists were already mature materials and the modifications needed for immersion lithography were rather minor. Only few groups worldwide were active in the resist field and often the efforts were diverted in different directions. On the other hand a significant research direction for lithographic materials followed by many polymer groups was toward self-assembled block copolymers for directed self-assembling (DSA) lithography [1,2,3,4]. The momentum seems to have changed during the last five years after the choice of EUV as the next industrial technology became clear in the lithography community along with the now well-recognized demand for high sensitivity EUV resists.

In the following we will try to make clear why the demand for high sensitivity resists causes a mobilization of the lithography researchers and brings again the resist development effort at a spot of increased interest by many scientists or engineers new to the field.

## 3. Material Absorption at EUV

The absorption of resist materials at EUV is an issue that only lately has started to attract the attention it deserves [29,30]. From an historical point of view, it should be mentioned that the resist absorbance at EUV was first considered at late nineties and it was recognized that the absorption of resist materials at EUV is mainly determined by their atomic composition. Indeed, the photon energy at 13.5 nm is close to 92 eV which is well above the ionization potential of the atoms that constitute the resist material. The photoemission cross sections of different atoms that can be potentially included in a resist composition are shown in the graph below, Figure 2, image adopted from ref [30] based on data from ref [31].

The atoms encountered in a typical resist composition are carbon, oxygen, and hydrogen. As shown in the graph of Figure 2, oxygen has a much higher photoemission cross section at EUV than C, thus increased the ratio of oxygen atoms in the composition is expected to increase the material absorption of 92 eV radiation resulting to enhanced electron emission [30]. On the other hand, one can further control the material absorption at EUV by incorporating molecules containing other atoms such as Hf, Zn, Sn, some of which are considered for the first time, in the resist composition.

In the first years of investigating options for resist materials suitable for exposure at 13.5 nm it was realized that high material absorption was not an issue of concern at EUV as was for 193 nm and 157 nm lithography [32]. On the contrary, it was soon noticed that the typical organic resist materials had rather low absorbance at the thicknesses of interest and this low absorbance would also result in low sensitivity. Thus, the incorporation of atoms in the resist composition that could substantially increase the materials absorbance was proposed among other strategies to increase sensitivity [17,18]. According to the data presented in Figure 2 such atoms could be Sn and I, compounds of which are indeed under investigation as the main components of highly sensitive resists and they will be discussed below.

In a deeper examination of the facts related to the increase of the resists absorbance one should consider the probabilities of absorption exhibited by the different atoms and correlate them with data for the absorption cross section for different wavelengths of ionization radiation. Relevant data can be also found in ref [33] where the different atom cross sections at a range of radiation energies are presented and correlated with specific atomic energy levels. By examining the tables for different atoms, it is clear that the main tendency is that the ionizing radiation interacts preferentially with electrons lying at deeper levels if they are at a distance smaller than 92 eV from the vacuum level. Thus, although for the case of C electrons can be released only from the 2s and 2p levels, for heavier elements there are core electrons at deeper levels which interact preferentially with the radiation. In ref [29] a recent study is reported where molecules of similar composition are compared (see Figure 3). There, it is shown that the substitution of H with e.g., halogen atoms has a dramatic influence on the formation of photoelectrons. The most interesting case is the one of iodine-substituted compound where the photoelectron spectrum intensity is much higher than that of the corresponding molecules containing lower atomic number atoms. This high intensity is due to the existence of core levels at favorable energy position and results in the formation of photoelectrons with rather low electron energies. The understanding of the impact of low energy electrons is a significant issue in designing efficient EUV resists. This point is under investigation in different resist systems, see for instance the case presented below in Figure 9.

One more remark that should be made here is that according to the discussion in ref [30] and the tables of references [31] and [33] it appears that it is quite improbable to have interaction of the incoming 92 eV photon with the electrons participating in the chemical bonds since these are substantially closer to the vacuum level. This remark rationalizes the hypothesis that the absorbance at EUV is mainly controlled by the atomic composition of the materials and not by the specific molecules used. Nevertheless, a deeper examination of the issue still has to be undertaken since the above considerations are usually based on energy position arguments and a detailed examination of the different electron states and their possible interactions with electromagnetic radiation has been undertaken only in very few cases.

## 4. Chemical Directions for Highly Sensitive Resists at EUV

General Considerations

Although the photochemistry involved in 248 and 193 nm is fully studied and the basic events occurring in those wavelengths are well-known, little work has been performed regarding organic material exposure at 13.5 nm. Actually, the chemical roots activated by the ionizing electromagnetic radiation have been so far considered mostly in very general terms not only for EUV but also for other wavelength ranges of ionizing electromagnetic radiation and even for different types of ionizing radiations including electron beams. The main reason for this fact is the difficulty encountered to identify and follow the plethora of possible events that are possible in specific organic chemical systems. Indeed, the ionizing electromagnetic radiation mainly gives rise to the formation of free electrons that can transfer different amounts of energy to the molecules through a number of interactions with the bound electrons. Nevertheless, in many cases the prevailing chemical outcomes are due to the availability of efficient chemical routes in the material, triggered by the formation of active intermediates like certain radicals, acids, or bases that act as initiators or catalysts. Some examples of such routes are the polymer degradation encountered in poly(meth) acrylates and related back-bone breakable polymers or the acid catalyzed deprotection or crosslinking reactions encountered in chemical systems best known as chemically amplified resists.

Such chemical reactions were discussed in X-ray lithography resists investigated until early 2000s when the EUV lithography appeared as an attractive option [34]. X-ray lithography based on radiation with wavelengths in the 0.1–10 nm range had been broadly investigated since these small wavelengths were considered attractive for high resolution applications. Nevertheless, the lack of good imaging technologies for such radiation had led to the adoption of the proximity exposure option which limited the possible gain from using small wavelength. The resists mainly explored during this period had been adaptations of UV or e-beam resists. The chemistries for solubility change observed were similar to the ones encountered in UV region especially for the case of poly(meth)acylate-based polymer resists such as PMMA (poly methyl methacrylate) and also for the acid-catalyzed systems (chemically amplified resists). The sensitivity of such resists in these wavelengths was not investigated in depth and the primary events after the absorption of electromagnetic radiation were not elucidated. The above approach was also adopted in the first resist efforts in EUV (13.5 nm) and the behavior of resists was compared with results obtained with different types of ionizing radiation as in ref [34]. A review of 2010 presents in a detailed manner radiation chemistry issues especially for the case of chemically amplified resists [35].

During the past decade the need for increased sensitivity which proved to be a significant requirement for the EUV lithography to become a competent industrial option ignited the research effort for especially designed sensitive and high-performing materials. The development and optimization of such high sensitivity resists pushed the scientific community to investigate in depth the radiation induced events and initiated a very demanding research area for new types of lithographic resists and for understanding fundamental issues [36]. In this respect, since photoelectrons and secondary electrons resulting from ionizing radiation were recognized to play the main role in solubility changing reactions [37], attention was drawn to the fundamentals of electron-induced chemistry known from other scientific fields as described for instance in [38,39]. Recently a study on the role of low energy electrons in a tin containing resist was published [40] and revealed their important role in resist chemistry. Similar studies are expected to dramatically improve the understanding of the resist chemistries at EUV.

In the following sections the different material options under investigation are discussed. These options include the introduction of unconventional materials such as the metal oxide containing resists and the in-depth understanding and optimization of the radiation chemistry encountered in more traditional resists as the back bone breakable polymers and the chemically amplified systems.

### 4.1. Chemically Amplified Resists

The imaging chemistry of the industrial resists used in 248 nm and 193 nm is based on an acid-catalyzed mechanism known as “Chemical Amplification.” The chemical amplified resists (CARs) are mainly consisted of a main polymeric or molecular component as a matrix, photoacid generator molecules, and base quencher molecules. Upon irradiation, the photo-acid generator molecules (PAGs) interact with the light to generate acids; then the change in dissolution rate of the matrix begins during a subsequent post-exposure bake (PEB) step. During this step reactions of specific groups of the matrix are catalyzed by the photogenerated acid molecules changing hydrophilicity, or inducing crosslinking or back bone scission, and hence alter solubility, usually in aqueous base developers. The design of the resists used for 248 nm and 193 nm relied on the tune of the absorptivity of the matrix, the type and size of the molecular components in the matrix, the expected etch resistance during pattern transfer as well as on several additional properties which controlled the performance of the resist [4]. Nevertheless, no such absorption selectivity of the main components of the resist materials can be expected when an ionizing radiation such as EUV (13.5 nm) is used. In fact, traditional CAR materials are very transparent in this wavelength, a fact that constitutes a major problem since it reduces the sensitivity of these materials. For this reason, new materials consisting of atoms with enhanced absorptivity in EUV regime have been proposed as alternative to traditional CARs. As discussed above the interaction of EUV photons with matter is mediated by photoelectrons, as well as secondary electrons. The broad energy distribution of these electrons that induce chemical changes in the material results in a variety of chemical events. The processes involved are quite complex and hard to disentangle, as photon illumination initiates often an entire electron cascade and the possibility of discerning the role of electrons with different energies is in many cases practically impossible. Consequently, it is not surprising that also in the case of CARs, as in other material cases, the interaction of low energy electrons (<15 eV) with soft matter is not well understood.

Toward improving our understanding on the behavior of CAR materials at EUV new experimental approaches have been proposed. Pollentier et al. in their paper [41] proposed an experimental approach based on residual gas analysis (RGA) to distinguish between photogenerated acid related reactions and direct scission reactions in model resist systems. In a more recent paper [42], also by Polentier et al., the residual gas analysis (RGA) was optimized and this method was successfully used to quantify the photogenerated acid yield and the reactions leading to insolubility as a function of EUV dose for a number of CAR formulations related to a commercial material.

On the other hand, new PAGs are introduced and are expected to perform more effectively at EUV compared to traditional PAGs. In a characteristic paper by Torti et al. [41], new fluorinated aryl sulfonates were investigated as initiators for EUV lithography and compared with traditional PAGs in epoxy-based hybrid materials. In Figure 4 the new proposed fluorinated PAGs are shown (upper row) along with the traditional PAGs which were used for comparison. As expected according to the atomic cross sections presented in Figure 2, the use of fluorinated PAGs resulted in resist formulations of enhanced sensitivity.

In the next section, we review the CAR materials used nowadays in EUVL as well as the proposed ways to improve their performance.

#### 4.1.1. Polymeric Systems Based on Hydrophilicity Change

Polymeric chemically amplified resists have been deeply investigated and look most established at this point. Traditional 248 nm and 193 nm photoresists based on acid catalyzed hydrophilicity change of polymer pendant groups were first evaluated in EUV lithography. Typically, photoacid generator (PAG) molecules are applied as a source of acid catalyst triggered by EUV exposure. The PAG components can exist in the resist formulation as individual moieties or as incorporated groups in the resins (polymer bound PAGs). In this section, recent progress in the design of these materials following various strategies is described.

In an early study, Yamamoto et al. used polyhydroxystyrene and partially protected polyhydroxystyrene with tert-butyl and adamantly groups to examine the dependence of acid generation efficiency on the protection ratio of hydroxy groups in chemically amplified EUV resists. Their study showed that protection of hydroxy groups clearly affects the acid generation process. The incorporation of t-butyl groups decreased the acid generation efficiency while adamanthoxyethyl groups increased this efficiency [44]. As has been mentioned above the primitive efforts for development of resists for EUV lithography followed the deep knowledge that had been acquired from the mature resists used in longer wavelengths. The first evaluated resists in EUV lithography were simply extensions of the commercially available previously developed environmentally stable chemical amplification photoresists (ESCAP) consisting of poly(p-hydroxy)styrene (PHS)/styrene/t-butyl acrylate copolymers as they are presented in Figure 5.

Fedynyshyn et al. reported a study on ESCAP resist using an EUV illumination source which revealed the effect of the matrix on the acid generation efficiency. They found out that while the absorbance was considered to be the important parameter, other polymer properties also strongly influenced the acid generation efficiency of the used PAG. The nature of the polymeric matrix, i.e., the different atoms and chemical groups that are present, plays an important role in the acid efficiency and has to be taken into account in order to achieve effective sensitization on the PAG. It should be noticed that these specific resists have well-defined processes and therefore they were extensively used in the early development of exposure tools. They showed capability for 30 nm half-pitch (hp) resolution with the required sensitivity, but it became clear that new designed resist materials optimized for the 13.5 nm wavelength exposure were needed for further performance improvement [45]. Tamaoki et al. compared polymer-bound PAG and polymer-blended PAG type resist materials in terms of blur, swelling properties, and lithographic performance. They found out that the polymer with bound PAG acquired very small blur with higher sensitivity and suppressed swelling very well [46]. Tarutani et al. investigated the effect of the hydrophobicity on the ultimate resolution of a photoresist material. Accordingly, they synthesized a series of polymers in which their hydrophobicity was changed by utilizing polymers having a different chemical structure and protection ratio. The polymer with the higher hydrophobicity can resolve 16 nm hp line/spacer using an EUV tool [47]. In another work of the same group, CARs with different sensitivities were synthesized in order to examine the impact of sensitivity on 15 nm hp resolution. Their results suggested that there was a strong relation between the exposure dose and the quality of the created structures. The resist with lower sensitivity (>30 mJ/cm^2^) could resolve 15 nm hp because of the low impact of photon shot noise [48]. Other issues concerning EUV resists include Out of Band (OoB) radiation which deteriorates the resist performance. OoB is estimated to be about 4% of the radiation from the EUV tool. PAGs with selectivity to EUV radiation have been designed and synthesized to minimize the effect of OoB in resist performance. These PAGs exhibited decreased Deep UV (DUV) absorption by the incorporation of insensitive cations and they were considered in resist formulations based both on blended PAG and polymer-bound PAG. The concept and merit of OoB insensitivity was confirmed by the obtained DUV and EUV sensitivity [49,50,51,52].

Liu et al. found out a chemical way to introduce PAG moieties in polymeric chains. This direct modification of polymeric chains was based on the introduction of sulfonium chloride onto the benzene ring of PHS by a convenient direct reaction at a high rate and then the anion was exchanged into perfluroalkyl sulfonate [53]. Narasimhan et al. studied the interaction between electrons having similar energies to secondary electrons produced during the EUV exposures and investigated resist materials using both experimentation and modeling [54]. JSR corporation developed new CAR EUV resist formulations showing short acid diffusion length as well as new sensitizers with higher EUV photo absorption atoms. These resist formulations showed capability to resolve 13 nm hp. Furthermore, addition of a new sensitizer to conventional CARs can improve the sensitivity about 9–16% with no affection on resolution and line width roughness (LWR) [55,56]. Krysak et al. established a method of pattern collapse mitigation in CARs using a dry develop rise material. This method is able to extend the resolution limit of chemical amplified resist susceptible to pattern collapse, resolving 24 nm pitch features [57]. Thackeray et al. found out that resists with low activation energies of deprotection can achieve superior process window and exposure latitude in the 35 nm resolution regime. In addition, they use photo-destroyable quenchers to minimize the loss of the photogenerated protons [58,59]. Fujii et al. improved resist sensitivity by increasing the proton source content in the polymer and applying an electron withdrawing group on PAG cation. They fabricated 13 nm hp line/spaces using newly developed chemical amplified resist materials combining the aforementioned items for enhancing the acid generation efficiency and suppressing the acid diffusion length [60]. Yamamoto et al. improved the sensitivity of chemically amplified resist by adding a metal sensitizer. The improvement in sensitivity is not a result of higher EUV photon absorption but of higher acid yield and electron efficiency. They achieved 43% improvement in sensitivity as well as reduction in LWR [61]. Fallica et al. measured the rate of bleaching by tracking the change in absorptivity of CARs during exposure to EUV light. They found out that the bleaching speed depended on the PAG-polymer interaction. This is an important effort in resist design and development because of the fact that Dill C parameter can be tuned in a variety of ways [62]. Lee et al. developed a multiscale model for EUV patterning of CARs. This model gives insight information about the chemical reactions (diffusion, quenching, deprotection etc.,) taking place during the structuring. Furthermore, it can predict the polymer loss during PEB as well as LER performance [63].

Additional ways to improve chemical amplified resist performance have also been proposed. Brainard et al. developed and evaluated eleven acid amplifiers for use in EUV photoresist. Acid amplifier (AA) is a compound that decomposed rapidly after the influence of an acid to generate more acid. They used an ESCAP photoresist measuring the performance after the addition of an AA. They found out that AA producing fluorinated sulfonic acids shows great promise in helping EUV resists, simultaneously improving the resolution, LER, and sensitivity [64]. Sekigushi et al. studied how the addition of metal into an ESCAP resist influences the sensitivity. They performed transmittance measurements and sensitivity evaluation of an ESCAP type resist doped with ZrO_2_ and TeO_2_ nanoparticles which have low and high absorptivity in EUV radiation respectively. ZrO_2_ nanoparticles caused no change in absorption and only slight sensitivity enhancement, whereas TeO_2_ nanoparticles enhanced both absorption and sensitivity [65]. Jiang et al. compared the impact of metal salt sensitizers and halogenated sensitizers on EUV sensitivity. Metal sensitizers improve both EUV photon absorption and electron yield resulting in higher sensitivity. Fluorine and iodine sensitizers also improved electron generation with their higher absorption but the chemical environment where these halogens are bonded influences heavily the sensitivity [66]. Nagahara et al. introduced Flood exposure Assisted Chemical gradient Enhancement Technology (FACET) to improve the resolution, process control, roughness, patterning failure, and sensitivity in EUV resist. Their concept is based on the increase in UV absorption after the influence of a EUV generated acid. After that the UV flood exposure induces acid production in these areas [67]. Okamoto et al. investigated the effect of the addition of diphenyl sulfones into EUV CARs. They confirmed that the addition of sulfones significantly increases the acid yield leading to the increase of the sensitivity of the CARs [68].

#### 4.1.2. Polymeric Systems Based on Acid-Catalyzed Main Chain Scission

Chemically amplified resists that can undergo acid catalyzed chain scission have been proposed as an alternative to resists based on hydrophilicity change in order to achieve high sensitivity at EUV. In these resists the imaging chemistry is based on acid-catalyzed back bone breaking instead of the deprotection of a pendant group. The removable units are attached at low molecular weight monomeric units instead of polymeric chains. Cardineau et al. synthesized polymers containing either tertiary aliphatic or tertiary benzylic cleavable ethers. But, further development would be made to overcome damage drawbacks such as serpentine pattern deformation and bridging [69]. In a similar approach, Manouras et al. designed and synthesized a random copolymer containing acid-cleavable bonds along the main chain. As it is presented in Figure 6, the random copolymer consisted of three different monomers corresponding at percentages of approximately 85%, 10%, and 5%, connected with acid labile bonds. Each monomer introduces or tunes a specific property of the random copolymer such as etch resistance, EUV absorptivity, Tg etc. Obviously, the bond strength of the monomer at high percentage dominated the polymer back bone breaking. Resist films based on the synthesized polymer have shown satisfactory etch resistance, due to the high aromatic moiety content. The sensitivity of this polymer is very high and the high values of Tg maintain the polymeric thin film untouched in the exposure step. The main chain of the polymer steadily chopped in the post exposure bake step in which the photogenerated acid can easily penetrate into the polymeric matrix. Resist formulations based on the synthesized random copolymer were exposed to EUV radiation exhibiting a high potential for industrial applications. A characteristic contrast curve shows that the resists are ultra-high sensitive (~0.5 mJ/cm^2^ using 5% PAG and 0.25% quencher) with satisfactory contrast, whereas they also showed very good etch resistance (1/10 selectivity to SiO_2_). Imaging experiments using EUV lithography have demonstrated capability for 20 nm lines with ultra-low doses <4 mJ/cm^2^ using 2.5% PAG and 0.15% quencher [70].

#### 4.1.3. Molecular Chemically Amplified Systems

Molecular type resists have attracted much attention for years since the small size of basic matrix material and its well-defined molecular structure are expected to be beneficial to better resolution and lower LWR. Past developments in the field of positive molecular resists including EUV materials had been reviewed in 2016 [71]. Currently, material’s robustness is often discussed like Tg, modulus, and adhesion to substrate along with efforts for high sensitivity. Echigo et al. developed a new molecular photoresist based on calixarene chemistry (Figure 7b). This resist has excellent solubility in conventional resist solvents and can be developed with standard alkaline developer TMAH. EUV patterning results showed resolution capability of 45 nm line and space as an EUV dose of 10.3 mJ/cm^2^ [72]. Figure 7 contains the precursor molecules of Noria photoresists (Figure 7a) and calixarene-based photoresists (Figure 7b). The hydroxyl groups contained in the aforementioned molecules can easily be modified producing photoresists with improved properties. Owada et al. prepared cyclic low molecular (CLM) weight resists with different protecting number. CLM resist achieved resolution of sub 30 nm hp patterns with high sensitivity [73]. Kudo et al. synthesized Noria derivatives with pendant adalantyl ester groups. They created 25 nm resolution pattern using EUV lithography. These patterns were obtained with less than 10 mJ/cm^2^ irradiation dose [74]. Kulshreshtha et al. synthesized a negative tone chemically amplified molecular resist based on modified Noria molecule having oxetane crosslinking moieties. Optimization of crosslinking can improve the balance between sensitivity, LER, and swelling. They have patterned 1:1 line structures with 20 nm resolution and 3.2 nm LER [75]. Another work of the same group demonstrated a blended resist system with higher performance by combining enthalpic and entropic contributions to solubility contrast. These resists have shown significant advancements in resolution, LER, and processability [76]. Dow electronics designed and synthesized several molecular glasses (MG) resists based on calixarene cores as alternatives to polymeric resists. They studied the relationship between the structure and the properties of MG resists in order to improve the lithographic performance. They created patterns with 28 nm 1:1 space lines using EUV exposure [77]. Frommhold et al. developed a new molecular resist system that showed high resolution capability. They optimized the performance of this system at 14 nm hp by 50% using a new quencher. Furthermore, dose improvements up to 60% was observed using metals as additives [78,79]. Irresistible materials Ltd. developed negative resist materials based on a multi-trigger concept. In a multi-trigger material, a catalytic process is utilized following the resist exposure in a similar manner to a chemical-amplified photoresist. However, in multi-trigger resist, multiple photoacids activate multiple acid sensitive molecules, which then react with each other to cause a single resist event. Figure 7c depicts the basic components of the multi-trigger resist formulations which are a molecular resin and an epoxy crosslinker molecule. Instead of a photoacid causing a single resist chemistry event as occurred in traditional CARs, in the case of multi-trigger resist concept, the photogenerated acid is being regenerated. This concept enables a high sensitivity solubility change above a certain dose threshold, but turns the resist respond off at lower dosages. This behavior is expected to lead to sharper lines and lower LER. Several studies have been performed to tune the quencher loading, metal addition, and resist design. Improvements on the design of these materials led to some excellent and very promising results in resolution, LWR-LER, and sensitivity. Resist formulations based on the multi-trigger concept showed capability to resolve 13.3 nm lines on 28 nm pitch, with 2.97 nm LWR and dose of 26 mJ/cm^2^ as well as 14.7 nm lines on 30 nm pitch, with 2.72 nm LWR and dose of 34 mJ/cm^2^ [80,81,82,83,84].

### 4.2. Non Chemically Amplified Photoresists

Development of non-chemically amplified resists has been proposed as an alternative to chemically amplified because of acid diffusion which may limit resolution improvement and worsen LWR for 22 nm HP and beyond. A key example is the excellent resolution and LER performance of the chain-scission polymethylmethacrylate (PMMA) resists. The chain scission reactions in these systems are well-known from the early days of Semiconductor Lithography [86], and the relevant reactions have been studied extensively. The backbone of the PMMA resist is cleaved under UV, ionizing radiation and electron beam exposures as shown in Figure 8, in the scheme adopted from ref [87]. Despite some excellent characteristics, PMMA resists have a number of drawbacks that have prevented their widespread use, such as the need for organic solvent development, high outgassing, poor etch resistance, and poor sensitivity. Different approaches are proposed to address these drawbacks and even recently it was a reported a new approach for enhancing PMMA pattern transfer performance through an infiltration process resulting in the formation of an AlOx film on top of the PMMA resist film [88]. This approach has been demonstrated so far with e-beam exposure. On the other hand in an increasing number of papers the performance of PMMA and related materials at EUV is investigated. Fallica et al. compared the performance of three high resolution lithographic tools. They used EUV interference lithography (EUV-IL), electron beam lithography, and He ion beam lithography tools to evaluate PMMA and hydrogen silsesquioxane (HSQ) resists under the same conditions. EUV-IL is a technique capable to pattern large areas of dense features with good resolution. Electron beam lithography is effective to fabricate high resolution arbitrary patterns and He ion beam is a promising technique to create both isolated and dense patterns because of almost negligible backscattered electrons [89]. ZEP resin has been extensively studied as an electron beam lithography resist material and has capability to provide high resolution patterns. The imaging enabling degradation mechanism is well studied and understood [90,91]. As it is shown in the Figure 8, PMMA and its derivatives undergo main chain scission when it is irradiated with proper radiation. ZEP520A was evaluated as a EUV resist using EUV-IL and exhibited better sensitivity in EUV radiation compared to PMMA. It yielded excellent dense arrays of 50 nm hp resolution and down to 25 nm hp with acceptable LER [92]. Sharma et al. developed a non-chemically amplified photoresist consisting of 4-(methacryloyloxy)phenyl dimethyl sulfonium triflate-r-isopropyl methacrylate. The resist has shown sensitivity of about 11.3 mJ/cm^2^ avoiding acid diffusion and blurring of resist pattern [93]. Oyama et al. introduced an easy method to predict EUV sensitivity by using electron beam sources. Considering that e-beam and EUV can induce the same chemical reactions, the required expose doses for the e-beam and EUV are expected to be related [94]. In a study at EUV [95] the influence of the PMMA molecular weight and processing parameters was examined. A characteristic EUV dose to clear was found to be 25 mJ/cm^2^ whereas crosslinking was observed at a dose of 600 mJ/cm^2^. The higher Mw material (950 K) gave the best printing results for 50 nm lines/spaces.

### 4.3. Inorganic Resists

In the first steps of EUV resist development, in an analogous way to the design of KrF or ArF resists, materials containing silicon attracted attention despite its low absorbing character to EUV light. However, realizing the importance of enhancing resist film absorbance to EUV light due to the low light source output at 13.5 nm, the silicon materials investigation as candidate EUV resists was substantially reduced. Alternatively, because of higher absorptivity in EUV radiation, resist materials containing metals were gaining interest. These materials are expected to have excellent sensitivity, better robustness and good etch resistance. Furthermore, it is expected to have improved properties concerning the relation among resolution, line edge roughness (in general, pattern quality), and sensitivity.

Most of the promising materials proposed are based on the inclusion of Hf, Zr, Zn, and Sn atoms in the resist formulation which are expected to enhance photoelectron production upon EUV radiation as expected from the data in Figure 2. In the following section we will review the promising materials that have been proposed trying to keep a chronological order, starting from the older ones. Since the imaging chemistries encountered are still under intense investigation, the recent papers focusing mostly on mechanistic issues will be discussed at the end of this topic on inorganic resists.

Ober’s group introduced metal oxide nanoparticles (NP) as next generation photoresist materials. The synthesis of these nanoparticles includes the controlled hydrolysis of zirconium or hafnium alkoxides in an excess of carboxylic acid followed by precipitation treatments to give ZrO_2_-NP or HfO_2_-NP with organic ligands. The size of the NP was controlled below 3 nm which is suitable for sub-20 nm lithography. Using either photoradical initiator or PAG, these NP gave both positive and negative tone patterns. These materials have shown high etch resistance as well as thermal and chemical stability. They have capability to resolve 26 nm lines using only 4.2 mJ/cm^2^ EUV dose. Furthermore, they made several efforts to understand the imaging mechanism of these hybrid photoresists [96,97,98].

Cardineau et al. studied the photolysis of tin clusters of the type [(RSn)_12_O_4_(OH)_6_]X_2_ using EUV radiation and explored these clusters as novel high-resolution photoresist materials. The photolysis of the organic ligand after the EUV irradiation activates the cluster leading to agglomeration and results in the observed negative-tone imaging. They have resolved dense line patterns with 18 nm dimension [99]. Passarelli et al. developed organometallic carboxylate compounds [R_n_M(O_2_(R’)_2_] as negative-tone EUV resists candidates. The imaging chemistry of such a resist is based on the polymerization of its acrylic substituents. This system has demonstrated exceptional sensitivity printing 35 nm dense lines with 5.6 mJ/cm^2^. Furthermore, they found out that among antimony, bismuth, tin and tellurium containing materials, the antimony incorporation provides the more sensitive resist while tellurium the least [100]. Sortland et al. investigated the photoreactivity of platinum and palladium mononuclear complexes. Despite the fact that many platinum and palladium complexes show little or no EUV sensitivity, they have found that metal carbonates (L_2_M(CO_3_) and metal oxalates (L_2_M(C_2_O_4_) (M is either Pt or Pd) are sensitive to EUV radiation. They demonstrated that the use of palladium as a core metal offers faster resists than the use of platinum [101]. Fugimori et al. developed a metal containing non chemically amplified resist material showing ultra-high sensitivity and capability to resolve 17 nm resolution features with 7 mJ/cm^2^ [102]. Li et al. prepared Hf-based photoresist materials with three different organic ligands by a sol-gel method. These resists have shown high sensitivity in EUV radiation as well as capability to create high resolution patterns [103]. Inpria corporation developed directly patternable, metal oxide hardmasks as robust, high resolution photoresists for EUV lithography. They have achieved 13 nm half-pitch at 35 mJ/cm^2^ and 11 nm hp with 1.7 nm LWR [104]. On the other hand, Hinsberg et al. proposed a numeric model describing the chemical and physical mechanisms governing pattern formation in metal oxide (MO_x_) EUV photoresists. They used experimental measurements to develop a quantitative representation of the chemical and physical state of the MO_x_ resist film at each step of the lithographic process [105]. Xu et al. prepared discrete nanometer scale zinc-based clusters and used them as resist materials for EUV lithography. These materials have shown capability to resolve 15 nm features [106]. Zang et al. reported the dual tone property of the tin-oxo cage [(BuSn)_12_O_14_(OH)_6_](OH)_2_ photoresist. This resist has shown a positive tone behavior when it is irradiated with low dose of EUV or E-beam and a negative tone behavior when it is irradiated at higher dosages [107]. Sitterly et al. investigated the photoreactivity of six organometallic complexes of the type ph_n_MX_2_ containing bismuth, antimony, and tellurium as metals and acetate or pivalate as ligands. They monitored the photodecomposition using mass spectroscopy when they were irradiated with EUV. They found that both the metal center and the carboxylate ligands have significant influence on the EUV photoreactivity of these compounds [108]. Rantala et al. developed novel EUV resists based on organohydrogensilsesquioxane. These materials worked as negative tone resists and have shown excellent etch selectivity and ability to form patterns by using industry standard TMAH development process. Furthermore, they exhibited low LWR (<2 nm) with sufficient sensitivity (40–60 mJ/cm^2^) [109,110]. Thakur et al. prepared Zn-based oxoclusters having trifluoroacetate (TFA) and methacrylate (MA) ligands. The Zn(MA)TFA photoresist displays appreciable sensitivity toward EUV radiation [111].

The mechanisms in all the above materials are still not adequately understood. Recently a study on hybrid HfO containing resists was reported by Mattson et al. [112]. In this study EUV-induced reactions were studied by using in situ IR spectroscopy of films irradiated by a variable energy electron gun and insights on the solubility changing mechanisms were obtained. Another study for HfO and ZrO materials was published by Wu et al. [113]. They applied different spectroscopic techniques and confirmed the higher sensitivity for a Hf-based material, as expected from its higher absorptivity. The role of carboxylate ligands in the resist crosslinking was also confirmed.

Lately, a lot of interest has been devoted to tin oxo cage materials. Haitjema et al. studied the chemical behavior observed in tin oxo cage materials under UV exposure to get clues that could also be of use for determining the EUV imaging chemistry [114]. Further insights on tin oxo cages chemistry at EUV was provided in the recent work by Bespalov where the influence of electron energy was studied [40]. The main results from this study are depicted in Figure 9. It was found that when the electron energy was under 2 eV incomplete material crosslinking was achieved. On the other hand for energies above 2 eV the material was crosslinked providing a denser final film.

The above mechanistic studies are now starting to help the elucidation of imaging mechanisms in the novel inorganic or hybrid component containing materials that are proposed as EUV resists. The continuation of this effort is expected to lead in designing high sensitivity EUV resists of advanced performance in the near future.

## 5. Summary and Outlook

The developments on the evolution of EUV resist technology during the past decade reviewed in the previous chapters, which have been actually intensified during the last few years, deserve the attention not only of the lithographic community but also of the broader nanomaterials community.

In the Table 1 below we summarize the performance results of the most promising resist materials reviewed. The materials are listed according to the design principle and the type of the main component in the resist composition. The performance indicators include the best resolution reported along with the corresponding dose, whereas data on the pattern quality i.e., LER-LWR (line edge roughness or line width roughness are provided, where available).

The main directions of the EUV resist technology evolution are also depicted in the graph of Figure 10.

Based on the reviewed ongoing efforts towards new resist materials we could emphasize the following points:

First, the demand for high performing resists according to industry standards for sub 20 nm nanostructures and also the demand of high sensitivity for securing acceptable technology throughput led to both, revisiting the fundamental issues and proposing new non-traditional material solutions. The ongoing resist field effort is expected to enable the successful adoption of EUV technology by the semiconductor industry.

Second, the new materials that have been proposed as resist candidates have incorporated many novelties. Especially interesting is the case of resists incorporating metal nanoclusters and related compounds. These compounds were selected because of their high absorbance of ionizing radiation, in particular EUV, but it is conceivable that they could absorb in other spectral regions as well and can be of use in a plethora of other radiation-related applications. The fact that well-defined nanostructures can be formed with such materials can also have important implications.

Third, electron-induced chemical changes that can be controlled and directed toward specific results are encountered in other fields of chemistry. Thus knowledge that can be acquired in the resist development effort can be eventually utilized for devising new routes in chemical synthesis. It can be also of great help in biology-related studies where very often the results of ionizing radiation should be investigated in depth.

It should be finally noticed here that open questions related to resist chemistry optimization remain and that the last developments can be viewed as the beginning rather than the end in the effort toward devising ionizing radiation sensitive materials of high performance.

## Figures and Tables

**Figure 1 nanomaterials-10-01593-f001:**
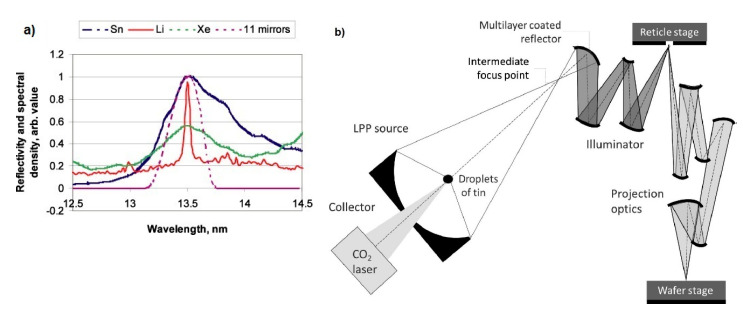
Extreme ultra violet (EUV) technology characteristics: (**a**) Emission of different sources based on Sn, Xe, and Li at the EUV spectral region and calculated near normal incidence reflectivity of a 11 mirror system in the same area. Adapted from [25], with permission from IOP Publishing, 2020 (**b**) A scheme of a EUV lithography system where the different parts, including source, illuminator, reticle stage (mask), and projection optics are depicted. Adapted from [17], with permission from De Gruyter, 2020.

**Figure 2 nanomaterials-10-01593-f002:**
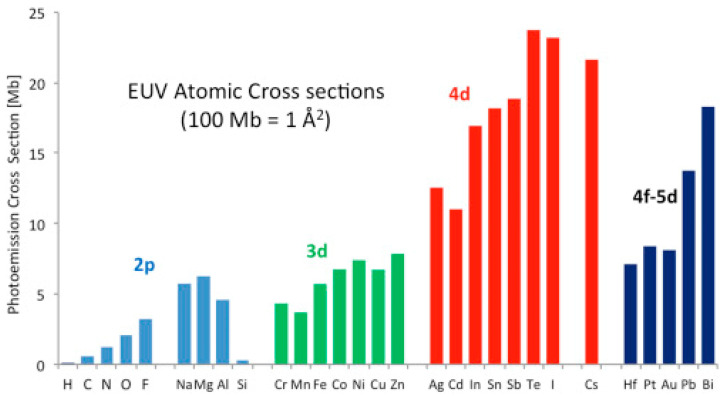
Photoemission cross sections at 92 eV calculated for selected atoms in Mb (Megabarn), where 1 Mb = 10^−22^ m^2^ in SI units. Figure adapted from [30], with permission from Elsevier, 2020.

**Figure 3 nanomaterials-10-01593-f003:**
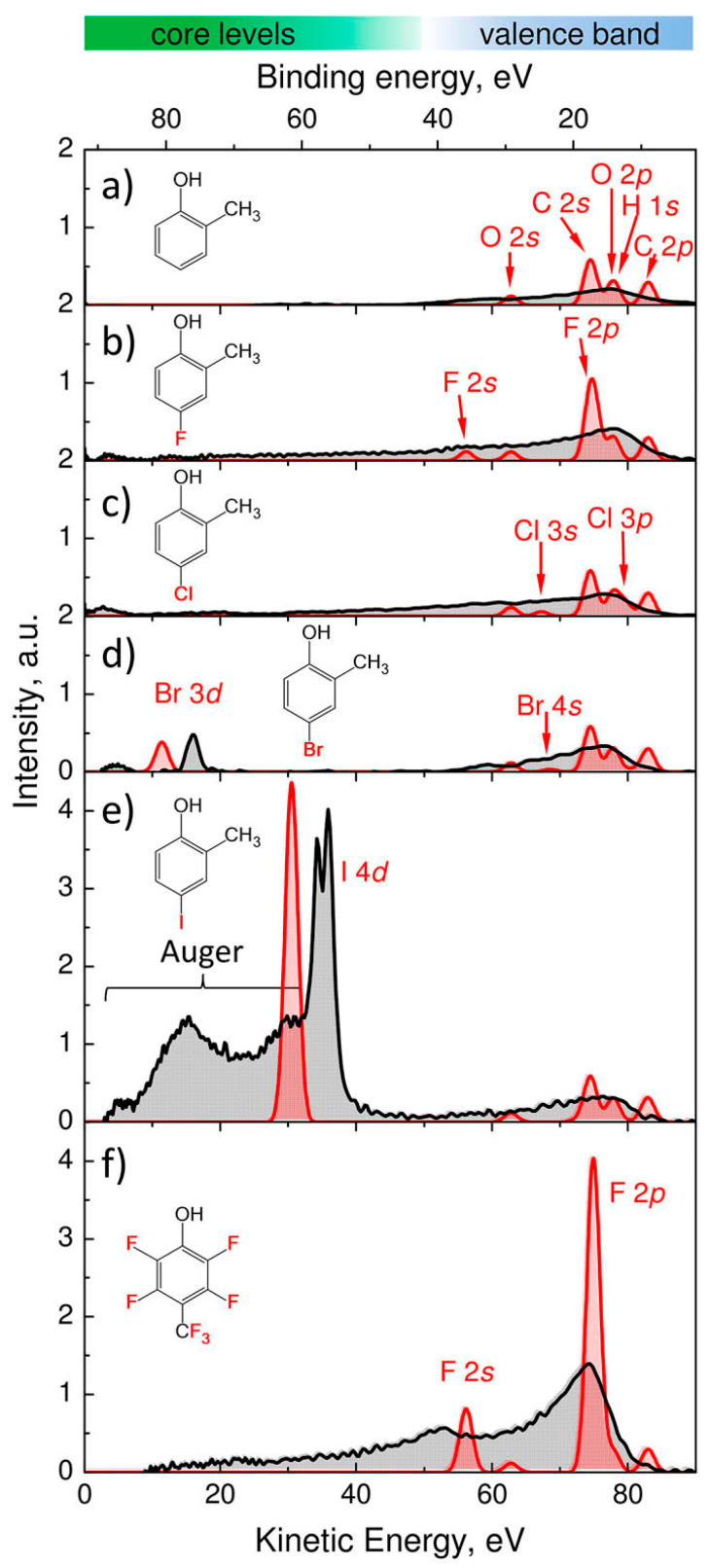
Photoelectron spectra of gas-phase molecules measured using 13.5 nm (92 eV) EUV radiation. Adapted from [29], with permission from AIP publishing, 2020. The kinetic energy of photoelectrons is shown in x axis. The black line corresponds to experimental data, and the red line corresponds to the model. (**a**) 2-methylphenol, (**b**) 4-fluoro-2-methylphenol, (**c**) 4-chloro-2-methylphenol, (**d**) 4-bromo-2-methylphenol, (**e**) 4-iodo-2-methylphenol, and (**f**) 2,3,5,6-tetrafluoro-4-(trifluoromethyl)phenol. The presented data clarify the importance of the inclusion of certain atoms in the resist composition. For instance, from the data presented it is clear that the inclusion of I in the resist composition is expected to greatly enhance the material absorption at EUV.

**Figure 4 nanomaterials-10-01593-f004:**
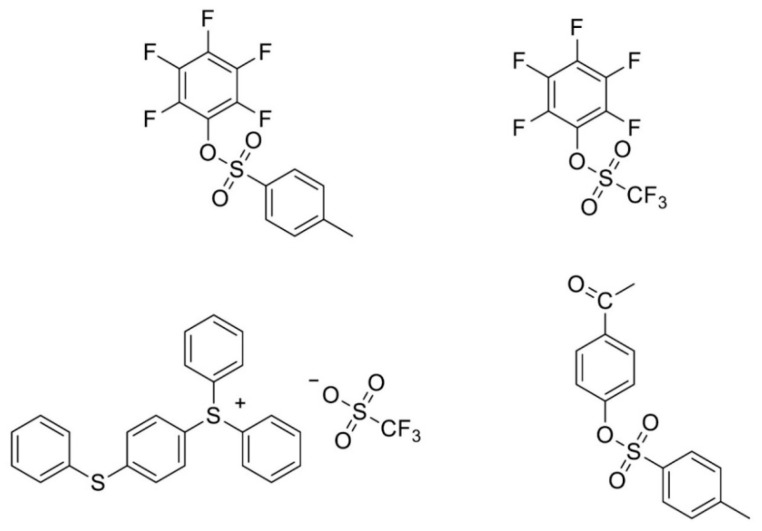
Fluorinated PAGs especially designed for EUV (**upper row**) and traditional PAGs (**lower row**) that were used for comparison in epoxy-based resist formulations. Adapted from [43], with permission from John Wiley & Sons, 2020.

**Figure 5 nanomaterials-10-01593-f005:**
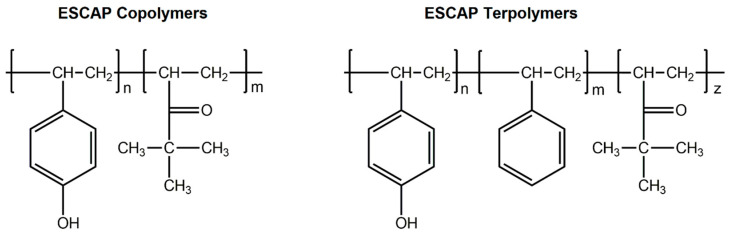
Polymers based on the environmentally stable chemical amplification photoresist (ESCAP) approach that are discussed in EUV resist formulations in ref [43]. The copolymer (**left**) consisted of poly(p-hydroxy styrene)-r-poly(t-butyl acrylate), the terpolymer (**right**) consisted of poly(p-hydroxy styrene)-r-poly(styrene)-r-poly(t-butyl acrylate).

**Figure 6 nanomaterials-10-01593-f006:**
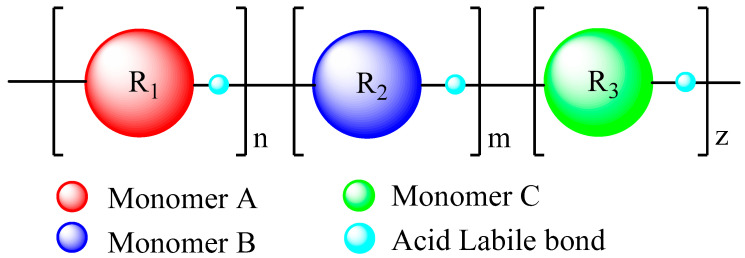
Chemical structure of the backbone breakable random copolymer. The copolymer is consisted of three different monomers connected by an acid labile bond. Each monomer introduces or tunes a specific property of the polymer. Adapted from [70], with permission from SPIE and the author Theodore Manouras, 2020.

**Figure 7 nanomaterials-10-01593-f007:**
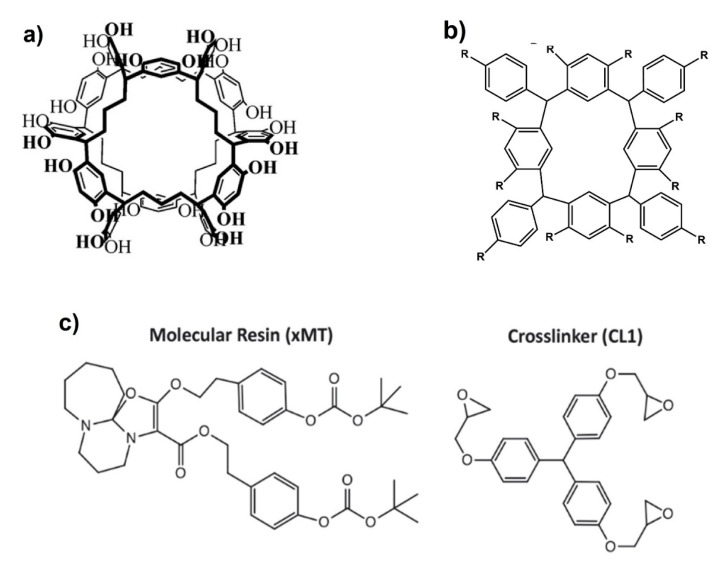
Examples of different cores for molecular glass resists that have been used for EUV lithography. (**a**) Noria molecule introduced in [85], graph adapted from [85], with permission from John Wiley & Sons, 2020, (**b**) a generic calixarene structure that has been the basis of a number of resist formulations and (**c**) basic components of a multi-triggered resist i.e., a molecular resin and a crosslinker, adapted from [83], with permission from The Society of Photopolymer Science and Technology, 2020.

**Figure 8 nanomaterials-10-01593-f008:**
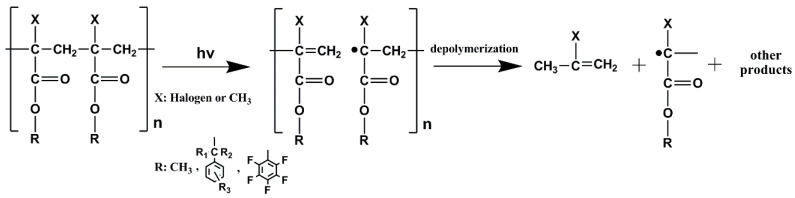
Example of a photodegradation mechanism for poly(methyl methacrylate) (PMMA) and its derivatives. Different atoms or groups strongly influence the properties of the corresponding resists including sensitivity and etch resistance. Irradiation of the polymer leads to the breaking of the main chain resulting to its depolymerization Adapted from [87], with permission from John Wiley & Sons, 2020.

**Figure 9 nanomaterials-10-01593-f009:**
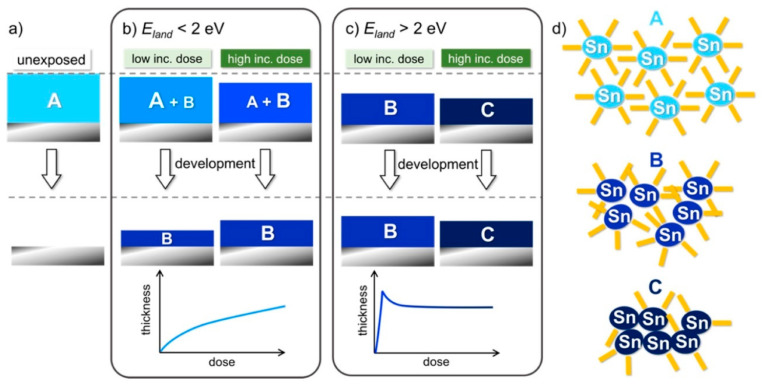
Scheme representing the patterning of the Tin-OH material. (**a**) The unexposed resist is removed completely during the development. (**b**) For electron exposure with E < 2 eV, only a small fraction of electrons impinging on the surface reach the material and only low conversion is attained. (**c**) For E > 2 eV, as the incident dose increases, consecutive reactions lead to the insoluble products B (denser than A) and C (denser than B). (**d**) Schematic representation of the initial Tin-OH molecular material A and of the two insoluble networks B and C. Blue ellipses represent the Sn-based inorganic core and orange bars the butyl chain. Adapted from [40] (https://pubs.acs.org/doi/10.1021/acsami.9b19004), with permission from American Chemical Society, 2020.

**Figure 10 nanomaterials-10-01593-f010:**
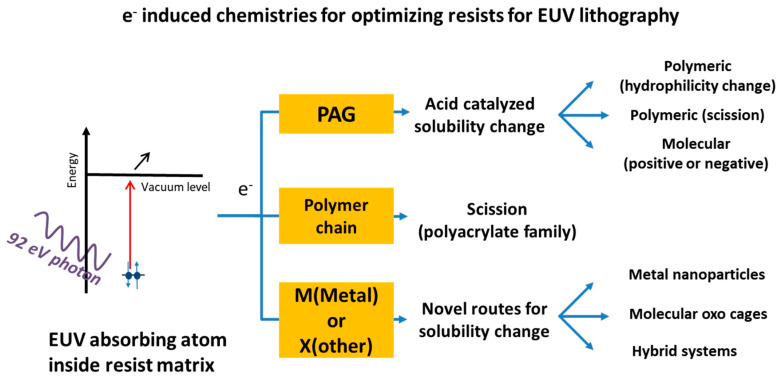
A summary of the most promising research directions that are currently explored toward high sensitivity and high performance EUV resists.

**Table 1 nanomaterials-10-01593-t001:** Performance characteristics (see text) of the most promising resists materials reviewed based on different design approaches.

Design Principle	Materials	Resolution	Sensitivity (Dose to Size)	Pattern Quality (LER-LWR)
CAR [45]	Polymeric	30 nm	<20 mJ/cm^2^	-
CAR [46]	Polymer bound PAG	24 nm	14 mJ/cm^2^	5.3 nm
CAR [47]	Polymer bound PAG-increased Hydrophobicity	16 nm	24 mJ/cm^2^	3 nm
CAR [48]	Polymeric	15 nm	25–30 mJ/cm^2^	6 nm
CAR [56]	Polymeric with different PAGs	13 nm	35.5 mJ/cm^2^	
CAR [59]	Polymeric	20 nm	31 mJ/cm^2^	-
CAR [60]	Polymeric	14 nm	43 mJ/cm^2^	5.8 nm
CAR [64]	Polymeric with Acid Amplifier (AA)	60 nm	1.9 mJ/cm^2^	7.9 nm
CAR-Multi-triggered resist [80,81,82,83]	Molecular	12.7 nm	53 mJ/cm^2^	4.2 nm
CAR [69]	Polymeric-main chain scission	20 nm	4 mJ/cm^2^	-
CAR [72]	Molecular	45 nm	10.3 mJ/cm^2^	-
CAR [73]	Molecular	45 nm	9.5 mJ/cm^2^	6.2 nm
CAR [74]	Molecular	26 nm	14.5 mJ/cm^2^	-
CAR [76]	Molecular	20 nm	40.5 mJ/cm^2^	3.2 nm
CAR [77]	Molecular	28 nm	22 mJ/cm^2^	3.7 nm
CAR [78,79]	Molecular	14 nm	36.1 mJ/cm^2^	3.26 nm
Non-CAR [93]	Polymeric	22 nm	78 mJ/cm^2^	<6 nm
Non-CAR [94]	Polymeric	20 nm	26.6 mJ/cm^2^	-
Non-CAR [95]	polymeric	50 nm	52 mJ/cm^2^	4.1 nm
Inorganic [97,98]	Nanoparticles	26 nm	4.2 mJ/cm^2^	-
Inorganic [101]	Clusters	18 nm	350 mJ/cm^2^	-
Organometallic [102]	Molecular	35 nm	5.6 mJ/cm^2^	-
Organometallic [103]	Complexes	30 nm	90 mJ/cm^2^	5.5 nm
Metal [104]	-	17 nm	7 mJ/cm^2^	5.6 nm
Metal oxide [105]	-	13 nm	35 mJ/cm^2^	-
Metal-organic [107]	Clusters	13 nm	35 mJ/cm^2^	-
Metal [108]	Complexes	50 nm	53.5 mJ/cm^2^	-
Organohydrogen silsesquioxane [110]	Molecule	22 nm	65.4 mJ/cm^2^	1.4 nm
Metal oxide [112]	Clusters	25 nm	37 mJ/cm^2^	-

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
