# Peer review of "High Sensitivity Resists for EUV Lithography: A Review of Material Design Strategies and Performance Results"

_nanomaterials, 2020, doi:10.3390/nano10081593_

Round 1

Reviewer 1 Report

Comments on manuscript nanomaterials 874716

„High sensitivity resists for EUV lithography: a review of material design strategies and performance results“

by T. Manouras and P. Argitis

Generally, such a review on materials for EUV is appropriate and serves to increase the understanding on material used for this new technology, which differs substantially in the technique applied, however, with respect to the materials the differences to optical lithography are not as distinct; common strategies are followed in parts, but also novel ideas are adequate and required to bring EUV lithography forwards.

In their manuscript of the review the authors have recapitulated a wide range of publications in this context; this is an excellent pool for researchers to make themselves familiar with this subject.

However, the authors should improve their manuscript with respect to what they propose in the title: “material design strategies”. The different strategies do not become as clear to the reader as they may be to the authors.

Furthermore, as a review is rather read by newcomers than by specialists, I propose the following changes:

General:

A) It is common to implement single figures from the publications reviewed. This is ok. However, as the figures are not explained in the text in a review (this would disrupt the presentation) they have to be explained, in short, in the respective figure caption. The figure caption should tell, what is shown in the figure, and include e.g. all abbreviations occurring there. In this respect improvement is required for almost all figures.

B) Some of the paragraphs are rather long. They should be broken down and provided with sub-topics so that the reader gets a better idea of the ordering principles followed by the authors – otherwise the goal of a review is missed. This is in particular the case with “chemically amplifies resists” in part 4. Choose 4., 4.1, 4.1.1 etc for the subtopics (or something accordingly that is common with the journal).

C) Each paragraph or sub-paragraph should start with 1-2 sentences concerning the principle (of material improvement or physics behind), before the manuscripts and their special results are listed, so that the reader is able to follow the order existing in the authors brain.

With these general improvements the review will be helpful for scientists, and (only then) it will be cited – I think, that the authors are interested in being “seen” by others.

There are, in addition, some more specific questions/issues:

Line 116/119: References 12 and 13 are confusing

Line 118: Indicate the transfer to Mb to SI-units!

Fig. 2: The ordinate indicates “Photoemission cross section”, but the authors talk about “absorptivity”. Add interdependence.

Line135/136: What is the difference between “probability of absorption” and “absorption cross section”? Add, e.g. in parentheses.

Line 148: add e.g. “… rather low electron energies (Fig. 3 e) that are of specific interest for development of highly sensitive EUV-resists”  - what is the reason that low energies are of impact?

Line 151/152: change to: “… the inclusion of e.g. halogen atoms in …photoelectrons is also indicated”

Fig. 4: most left (1) = not absorbed?? (remove (1)!). That (2) and (3) are important with EUV only should be explained in the caption. And: why is the “deprotection sphere” an ellipse? Has the figure been squeezed vertically? Or is that physical?

Line 248: Insert sub-topic and principle!

Fig. 5: give at least names of the three different monomers, as well as reason for their use (the matrix should…)

Line 261: Insert sub-topic – or does it belong to the paragraph before?

Line 264/265: add “role” of matrix – what is the intention followed?

Line 288: Insert sub-topic and principle!

Line 316: “PES” is not explained/defined!

Line 317: insert sub-topic and principle!

Line 334/335: What is the benefit with using multiple acid sensitive molecules, e.g. by an example!

Line 345: Insert sub-topic and principle!

Line 364: explain difference between/ sense behind one acid labile bond per monomer (and use of 3 different monomers). Are they cleaved at different energies???

Line 365: Molecular systems (In my view, we are still in the chapter on CAR resists…)

Line 367/368: What are the other components? Without solvent? Seem to be suitable for negative tone – what about positive tone? Explain principles followed!

Line 375: Noria = ?? abbreviation? Name?

Line 384/385: difference of “molecular glass” from other concepts? Or only different name? Add!

Line 392: Is Noria a modified Calixarene structure? If not, what is it?

Line 405: HSQ not introduced so far!

Line 421: Is the cleaving similarly efficient with the different “X” and/or “R”-groups? Add!

Line 424: “due” to its low.. or “despite” its low absorbing??

Line 430: RLS not introduced

Line 439: add short explanation: what is “O” (oxygen??), what is “A”

Fig. 9: Shift to a position later in the text, near to ref 73!

Lines 423-481: break down in sub-topics and add sub-titles! (in particular line 441-481) This paragraph is highly unclear and the order is not discernible for the reader!

Fig. 10: Explanation urgently required! Topics should correspond to issues/topics mentioned in the review (sub-titles).

Lines 491-499: Remove history – this is the summary of the review!

Additionally, some minor typos should be corrected, according to:

Line 46/47: “… power making the improvement in resist sensitivity a high importance issue to fulfil…”

Line 71:”…a technology that has reached…”

Line 90: “…remains relatively low…”

Line 104: “…along with the now well-recognized demand…”line 121: “…higher absorptivity than C, thus an increased…”

Line 128: “…also result in low…”

Line 140: “…distance smaller than 92 eV from…”

Line 148: “…and results in the formation…”

Line 257: “…(ESCAP) consisting of…”

Line 261: ”…resist using an EUV…”

Line 282: “…OoB…”

Line 313: “…Dill C parameter…”

Line 322/323: “…et al. studied how the addition of metal into an ESCAP resist influences…”

Line 332: ”… based on a multi-trigger…”

Line 383: “… resists have shown significant…”

Line 397: “…amplified because of acid diffusion…”

Line 404: “…(EUV -IL)…”

Line 465: “…representation of the chemical…”

Line 484: “…The developments on the evolution of EUV resist technology during the last decade reviewed in the previous chapters, which…”

Dear authors,

I am looking forward to a well-organized review which is clearly ordered and which indicates the main principles followed, so that it is of high value for a reader interested in this subject, and, in addition, which scientists enjoy to reference!

Reviewer 2 Report

In this review the authors briefly introduce why the semiconductor industry is adopting extreme ultraviolet lithography as the method of choice to fabricate the next generation of integrated circuits. They give a list of different works in which resist materials have been investigated for EUV lithography applications. They group them in two categories: chemically amplified resists (CARs) and non-chemically amplified (nCAR). Within the last group, the subgroups of polymer-based and inorganic/hybrid materials are given. The topic is relevant and the introduction is nicely written and puts the reader into context. Yet, the manuscript need quite some improvements before it can be published.

As indicated in the title, the authors go through different strategies that have been reported. However, they only do this with a long written list of examples in the literature, which, in fact had been mostly covered by previous reviews (Chem. Soc. Rev. 2017, 46, 4855-4866, cited, and RSC Adv. 2020, 10 (14), 8385–8395, NOT cited) I am missing items like tables or schemes to help the reader have a wholistic view on what is the actual situation and underlining what are the pros and cons of the different strategies. The authors are not really complying with the part of the title “performance results”. To do so, they could include a table with the best reported results for sensitivity (understood as dose to clear/gel) or dose to size), resolution (minimum critical dimension) and roughness for each type of resist that they mention (when available, of course). Right now, they only give some of these values for some of the resists.

I am also missing a more deep analysis on which are the critical points that have not been solved yet and why are all these strategies being pursued. Related to that point, the authors claim that CARs have low sensitivity (line 225) but this is not really true. They can print at very low dose. But they suffer from other problems related to resolution. The authors could also explain better the new concepts for polymeric resist specially designed for EUV. For instance, they do explain well the multi-trigger resist concept (line 333-336) and they do not even give the outstanding results that these materials are attaining. The same applies to the PS-CAR concept. Other very recent concepts, like infiltration of metals by vapor diffusion are not even mentioned (J. Mater. Chem. C, 2019, 7, 8803).

It would be also nice that they highlight which have been the findings in the last years regarding the EUV-induced chemistry in a more systematic way. They only show the work by Kostko and briefly mention Bespalov et al. to briefly discuss the role of low energy electrons, but they omit other dedicated works on electron-induced chemistry by Pollentier et al (Proc. SPIE 2018, 10586, 10560C or Proc. SPIE 2019, 10957, 109570I). These works are particularly relevant because they actually address the point raised by the authors in lines 231-232 (“the possibility of discerning the role of electrons with different energies is in many cases 231 practically impossible”)

In terms of contents, the authors left out other important works in the literature, like a recent review on inorganic EUV resists by Luo et al. (RSC Adv. 2020, 10 (14), 8385–8395). Also, the authors talked about scission mechanisms and use a study performed with electron beam instead of referring to studies on scission mechanisms induced by actual EUV in polymers made by Rathore et al. (J. Mater. Chem. C, 2020, 8, 5958) Similarly, Pollentier has published multiple works on electron- and EUV-induced chemistry on CARs, as mentioned above, but none of them are mentioned.

The part about inorganic resist should be better organized. The Sn-clusters published by Cardineau are the same studied by Zhang and Haitjema and yet they are not grouped together. Examples of Hf-based resist are also scattered through the paragraph. The authors should choose a criteria to talk about inorganic resist: do they group them by metal or do they group them by working principle? For instance, line 461, “Kosma et al. explained the mechanism of the imaging 461 chemistry of zirconium based hybrid photoresist materials”. And yet, not all Zr-containing resist will operate the same way because it mainly depends on the organic shell around them. Polymerization mechanisms for resist with acrylate types of inorganic resist (lines 445-447) have been thoroughly studied by Mattson (Chem. Mater. 2018, 30, 6192−6206 Article) and Wu (J. Micro/Nanolith. MEMS MOEMS 2019, 013504) but these references are missing.

Scheme 10 is a quite poor graph. The authors should branch out the graph to show all subgroups and all strategies currently being under study. I do not mean that they have to give ALL the resist, but, within CARs, there are many strategies: increase EUV absorption of polymer, or electron sensitivity of the PAG, the PS-CAR the multi-trigger, blending with metals and with PAG that contain metals... same for polymer chain or inorganic resist. They have different working principles that could be shown conceptually in this scheme.

Finally, the manuscript could be better written. Many sentences are missing prepositions (e.g. line 39, “IN the last two decates...”) or connectors (e.g. line 71, a technology THAT has reached an acceptable...). Grammatical errors (having combination (line 305), have designed (line 283)) and sentences that are not well written or are not appropriate for a scientific document like a review (“Of course, the photoacid is being regenerated” line 336). The authors should use hyphen in terms like “electron-induced (line 27)” “high-importance issue (line 46)”.

Reviewer 3 Report

The manuscript ‘High sensitivity resists for EUV lithography: a review of material design strategies and performance results’ by Theodore Manouras and Panagiotis Argitis is a review paper on Extreme Ultraviolet lithography, discussing state of the art resist materials for EUV and promising alternatives enabling highly sensitive EUV.

The article shows a broad overview of the EUV technique and on the available resist, mentioning the main challenges in the field. However, the comparison of EUV with alternative/ maskless lithographic approaches such as interference optical lithography (Lloyd scheme) or electron beam lithography is poorly discussed and referenced. I would suggest to the author to add a comment in this respect in the introduction in order to better highlight the pros and cons of the EUV with respect to other relevant approaches in nanotechnology.

Round 2

Reviewer 1 Report

The manuscript has improved substantially, thank you for your efforts!